# NLRP3 Inflammasome Inhibitors in Cardiovascular Diseases

**DOI:** 10.3390/molecules26040976

**Published:** 2021-02-12

**Authors:** Eleonora Mezzaroma, Antonio Abbate, Stefano Toldo

**Affiliations:** 1VCU Pauley Heart Center, Virginia Commonwealth University, Richmond, VA 23298, USA; emezzaroma@vcu.edu (E.M.); antonio.abbate@vcuhealth.org (A.A.); 2Pharmacotherapy and Outcomes Sciences, Virginia Commonwealth University, Richmond, VA 23298, USA

**Keywords:** NLRP3, inflammasome, caspase-1, ASC, IL-1, IL-18, cardiovascular disease, ischemia, heart failure, inhibitor

## Abstract

Virtually all types of cardiovascular diseases are associated with pathological activation of the innate immune system. The NACHT, leucine-rich repeat (LRR), and pyrin domain (PYD)-containing protein 3 (NLRP3) inflammasome is a protein complex that functions as a platform for rapid induction of the inflammatory response to infection or sterile injury. NLRP3 is an intracellular sensor that is sensitive to danger signals, such as ischemia and extracellular or intracellular alarmins during tissue injury. The NLRP3 inflammasome is regulated by the presence of damage-associated molecular patterns and initiates or amplifies inflammatory response through the production of interleukin-1β (IL-1β) and/or IL-18. NLRP3 activation regulates cell survival through the activity of caspase-1 and gasdermin-D. The development of NLRP3 inflammasome inhibitors has opened the possibility to targeting the deleterious effects of NLRP3. Here, we examine the scientific evidence supporting a role for NLRP3 and the effects of inhibitors in cardiovascular diseases.

## 1. Introduction

The innate immune system comprehends a genetically coded and inherited set of receptors that recognize pathogens based on the ligand chemical property. These receptors are therefore defined as “pattern recognition receptors” (PRRs) [1]. Several unrelated agonists sharing similar chemical properties can bind to the same PRR. Altogether, these agonists are referred to as “pathogen-associated molecular patterns” (PAMPs) [1]. During an infection, the activation of PRRs mediates the transcription and secretion of chemokines and cytokines, leading to activation and reprogramming of myeloid and lymphoid cells [2]. Due to the low selectivity of some PRRs, self-molecules released by damaged or stressed cells can activate PRRs in the absence of an ongoing infection. These molecules are altogether defined as “damage-associated molecular patterns” (DAMPs) and include several intracellular proteins that, when released outside the cell, act as “alarmins” (Figure 1) [1,2]. The production of DAMPs and the activation of PRRs following cell or tissue damage coordinate the process of damage resolution and healing [3]. Virtually every human chronic disease is associated with the activation of a set of PRRs [4,5,6].

Nucleotide-binding oligomerization domain (NOD)-like receptors (NLRs) are PRRs [7]. Several of the proteins in the NLR family form multiprotein complexes defined as “inflammasomes”, which lead to the activation of a pro-inflammatory enzyme caspase-1 (or caspase-11 in mice and caspase-4 and -5 in humans) and the secretion of cytokines of the interleukin (IL)-1 family [7]. NACHT, leucine-rich repeat (LRR), and pyrin domain (PYD)-containing protein 3 (NLRP3) is the most studied inflammasome due to its pathophysiological activation in infectious, rheumatologic, and chronic diseases [8]. Moreover, the NLRP3 inflammasome promotes tissue damage following acute organ injury, such as acute myocardial infarction (AMI) [9]. NLRP3 is a sensor of damage [7]. When active, it transforms the cell into a pro-inflammatory factory of cytokines and/or leads to cell death (Figure 1).

### 1.1. PRRs That Regulate Inflammasome Formation

NLRs include several families of cytoplasmic proteins, including NOD receptors; NACHT, leucine-rich repeat (LRR), and pyrin domain (PYD)-containing proteins (NLRPs); and NACHT, leucine-rich repeat (LRR), and caspase recruitment domain (CARD)-containing proteins (NLRCs) [10,11]. Some NLRs form inflammasomes, while others are signaling receptors. Nod1 and Nod2 are solely involved in inflammatory signaling through mitogen-activated protein kinases (MAPKs) and nuclear factor *kappa* B (NF-κB) [12,13]. NLRPs and NLRCs include some of the most widely studied inflammasome receptors, such as NLRP3, NLRP1, and NLRC4 [11,12]. NLRP3 has been extensively studied in the cardiovascular system [9,11,14]. NLRP1 has been shown to have a role in the response to viral and bacterial infections but to be less involved in sterile inflammation due to injury [10,12]. However, a recent study has found a homeostatic role of NLRP1 in controlling baseline levels of IL-18 [15]. In mice, NLRP1 deletion, like IL-18 deletion, promotes overeating, obesity, and glucose intolerance [15]. NLRC4 is highly expressed in the intestinal epithelium and is widely investigated in the interaction with intestinal bacterial flora [16].

The IFI20X/IFI16 family includes the non-NLR inflammasome forming receptor absent in melanoma 2 (AIM2) [17]. AIM2 is an intracellular PRR that identifies double-stranded DNA. When the DNA of replicating pathogens (e.g., viruses and bacteria) accumulates in the cytoplasm, AIM2 identifies the double-stranded DNA and initiates inflammasome assembly, leading to the activation of caspase-1, to the production of IL-1 family cytokines, and/or to cell death [18]. Although AIM2 is primarily responsible for the recognition of exogenous DNA originating from pathogens, it can detect mitochondrial DNA (mtDNA) [18].

Another class of PRRs important for the regulation of inflammasomes is the Toll-like receptors (TLRs) [19,20]. TLRs are type-I integral membrane receptors [21]. Their extracellular domain has several LRRs [21]. When an agonist binds, the extracellular domains of two TLRs get closer, forming homo- or hetero-dimers and leading to an interaction with their intracellular C-terminal Toll/interleukin-1 receptor (TIR) domain [21]. The TIR domain is conserved also in the receptors of the IL-1 family [9]. When activated, the TIR domains of the receptors interact with the TIR domains of the myeloid differentiation factor 88 MyD88, the TIR domain containing adaptor protein (TIRAP), the tumor necrosis factor (TNF) receptor associate factor 6 (TRAF-6), and the TIR domain-containing adapter-inducing interferon-β (TRIF), different intracellular adaptors needed for transduction of the receptor-activated signal [9,21]. The signaling of TLRs culminates with the activation of NF-κB, MAPKs, and/or proteins of the interferon-regulated transcription factors (IRFs) family [22]. As signaling proteins, the TLRs promote transcription of the inflammasome components and substrates (i.e., IL-1β and IL-18), a process termed inflammasome priming. Therefore, TLRs represent an important component of the mechanism of regulation of the inflammasome pathway by providing this priming signal (Figure 2) [23,24].

### 1.2. The NLRP3 Inflammasome

NLRP3 is a 118 KDa cytosolic protein that has three different domains: a LRR domain at the C-terminal, a nucleotide-binding and oligomerization domain (NOD) also known as NACHT, and a PYD at the N-terminal [23]. The LRR is the domain responsible for detecting the presence of microbial ligands and alarmins. The NACHT domain is important for NLRP3 oligomerization and has the active ATPase site through the Walker A motif (ATP-binding site) and the Walker B motif (ATPase activity) [23]. When the LRR domain is activated, NLRP3 monomers start to oligomerize through their NACHT domains (Figure 2) [9,10,11,12,13,14,15,16,17,18,19,20,21,22,23].

The active NLRP3 oligomerizes and binds to the adaptor protein ASC (apoptosis-associated speck-like protein containing a caspase recruitment domain or CARD) [9,25,26]. ASC contains a PYD domain, and the interaction with NLRP3 is mediated by the PYD–PYD interaction [25]. This leads to ASC polymerization with the formation of filamentous structures [25]. ASC then binds to the pro-caspase-1 interacting with the CARD of the latter [25]. This further promotes the growth of polymeric filamentous structures that, viewed with the electronic microscope, give the appearance of the inflammasome as a stellate structure, with the ASC polymers forming the “filaments” and the pro-caspase-1 polymers forming the “branches” [25]. Within these structures, pro-caspase-1 is activated by autocatalytic cleavage. The active caspase-1 in turn cleaves its substrates, which include the classical inflammasome cytokines pro-IL-1β and pro-IL-18, and the pore-forming protein Gasdermin-D (GSDMD) (Figure 2) [23,27]. The N-terminal mature form of GSDMD (NT-GSDMD) oligomerizes, leading to the formation of cell membrane pores that facilitate the extracellular release of the mature IL-1β and IL-18 [26]. These two potent pro-inflammatory cytokines are involved in the pathogenesis of several acute and chronic cardiovascular disorders (CVDs) [28]. In addition, the NLRP3 inflammasome drives a specific form of inflammatory-mediated cell death, called pyroptosis [9,27,28]. It is mediated by the activity of caspase-1 and the formation of the NT-GSDMD pore, which induce unregulated cellular ions movement through the cell membrane, water influx leading to cell swelling, and membrane rupture [9,27,28]. Caspase-1 can also shut down the glycolytic pathway by cleaving and inactivating key enzymes necessary for this process [29].

Under normal physiological conditions, the cytosolic levels of NLRP3, inflammasome components, and substrates are expressed at a low level even if NLRP3 is activated [30]. In this condition, the signal may be insufficient to induce polymerization and to reach the threshold necessary to form the inflammasome and to activate caspase-1. In fact, in many cells, the formation of the inflammasome is a “two steps” process [31,32,33]. A “priming” signal is necessary to stimulate the transcription of NLRP3 together with the inflammasome components and substrates, while a “triggering” signal leads to the activation of NLRP3 that culminates with inflammasome assembling and activation of caspase-1. The priming is mediated by signaling PRRs, including TLRs and Nod2, and cytokine receptors, which culminates with NF-κB-mediated transcriptional upregulation of the inflammasome components and its substrates (Figure 2) [19,20,34]. However, in the last few years, it has become apparent that other types of receptors (including tyrosine kinase receptors and G-protein coupled receptors), important in the regulation of cardiovascular system function, such as angiotensin receptor type 1 (AT1) or the adrenergic receptors, which are not usually involved in inflammatory pathways, can promote inflammasome priming (and triggering) (reviewed in [14]). More recently, posttranslational modifications, such as ubiquitination and phosphorylation, have been described to have a role in NLRP3 priming [35,36,37].

Several trigger signals have been identified for NLRP3 [9]. These signals are unrelated and include intracellular signals (e.g., lysosome rupture, mitochondrial dysfunction, and reactive oxygen species) and extracellular signals (e.g., ATP-mediated activation of the purinergic-type 2 receptor X7, P2X7, potassium efflux, and calcium influx).

## 2. Mechanisms of NLRP3 Activation in the Heart

The priming phase is mediated by several types of DAMPs. Following acute myocardial infarction (AMI), ischemia and the postischemic damage generate DAMPS and the intracellular and extracellular release of alarmins. Moreover, in additional cardiovascular diseases (e.g., atherosclerosis and hypertension) or in chronic disease associated with an increased risk of developing cardiovascular diseases (e.g., obesity or diabetes), priming is promoted by metabolites and/or neurohormonal activation (e.g., angiotensin II, fatty acids, and glucose) [38,39,40,41,42,43]. Priming during diabetes increases the expression of components and substrates of the NLRP3 inflammasome, inducing a condition that exacerbates the response to experimental AMI [44].

As stated above, several stimuli can contribute to the activation of NLRP3 (Figure 2). One of the most common triggers for NLRP3 activation is K^+^ efflux, which is regulated by several intracellular and extracellular proteins. The P2X7 receptor is a K^+^ channel leading to potassium efflux upon binding of extracellular ATP [45,46]. P2X7 inhibition or gene silencing following AMI reduces myocardial damage [47]. NEK7, a serine/threonine kinase member of mammal NIMA-related kinases, senses the change in intracellular K^+^ concentration and mediates the activation of NLRP3 [48]. Potassium efflux can also be induced by the leak of lysosomal content into the cytoplasm [49,50,51]. Crystals of monosodium urate, calcium phosphate, or cholesterol lead to dysfunctional phagocytosis, lysosome instability, and swelling, eventually causing lysosomal rupture and cathepsin B leakage, leading to potassium efflux and NLRP3 activation (Figure 2) [49,50,51,52,53].

Autophagy is an intracellular pathway that is responsible for the coordinated digestion of proteins and cytoplasmic organelles, including mitochondria [54]. In experimental settings, autophagy regulates the inflammasome pathway [55]. Impaired mitochondrial autophagy (mitophagy) can lead to an increase in reactive oxygen species (ROS) and cytosolic accumulation of mitochondrial DAMPS, including mitochondrial DNA (mtDNA) and cardiolipin (Figure 2) [56,57,58,59,60]. ROS and the intracellular redox balance are important signals leading to NLRP3 activation [56]. A protein sensitive to oxidative stress is the thioredoxin-interacting protein (TXNIP). TXNIP binds to the oxidoreductase thioredoxin (TRX) [61]. Under oxidative stress, TXNIP detaches from TRX and mediates NLRP3 activation [62]. TXNIP silencing before experimental AMI protects the heart [63].

## 3. Role of the NLRP3 Inflammasome in the Development of Cardiovascular Diseases

Over the last decade, the central role of the NLRP3 inflammasome in the development and progression of CVDs has been well established [9,14,28]. Atherosclerosis is associated with chronic inflammation and accumulation of lipids and macrophages in the plaque [64]. In macrophages, cholesterol crystals activate the NLRP3 inflammasome and promote IL-1β release [52,53]. The transfer of myeloid cells deficient in NLRP3 or ASC or in low-density lipoprotein receptor knockout mice (Ldlr^−/−^) recipient protects the mice from the development of atherosclerosis [53]. Inhibition of the NLRP3 inflammasome pathway reduces the infarct size and improves cardiac function in mouse models of AMI [47,63,65]. During AMI, prompt reperfusion of the tissue is needed to limit the tissue damage. However, the reestablishment of blood flow and oxygen supplementation to the ischemic tissue leads to so-called reperfusion injury, leading to cellular and tissue damage. Following ischemia-reperfusion, NLRP3 activation is delayed due to the priming signaling phase mediated by DAMPs and alarmins released by dying cells [66]. Moreover, these DAMPS lead to the recruitment of myeloid cells to the injury site, exacerbating the injury [67,68]. In mice, the deletion of NLRP3, ASC, or caspase-1 reduces infarct size and improves cardiac function [47,63,65,69].

NLRP3 inflammasome levels directly correlate with heart dysfunction, the levels of the N-terminal (NT) fragments of the pro-form of brain natriuretic peptide (BNP) (NT-proBNP, a marker of heart failure), and the rate of hospitalization in patients with dilated cardiomyopathy (DCM) [70]. Biopsies of DCM patients show increased cardiomyocyte pyroptosis [71].

NLRP3 inflammasome expression and function are elevated also in mice subjected to pressure overload due to transverse aortic constriction (TAC) or implanted with osmotic pumps to deliver hypertensive doses of angiotensin II (AngII) [72,73,74]. Inhibition or genetic deletion of NLRP3 improves heart remodeling and reduces the inflammatory and fibrotic processes [72,73,74,75,76].

A pathophysiological role of the NLRP3 inflammasome has been also shown in animal models of injury due to anticancer treatments (i.e., chemotherapy and radiation therapy), obesity and age-associated metabolic derangements, diabetic cardiomyopathy, pericarditis, myocarditis, and cardiac sarcoidosis [14,28]. The role of inflammasome inhibitors in preclinical models of these diseases is reviewed below.

## 4. NLRP3 Inflammasome as a Target for Pharmacological Inhibition

The NLRP3 inflammasome has been associated with several inflammatory diseases (acute and chronic) and genetic autoimmune syndromes [7,8,9,14]. For this reason, in the past few years, there has been an effort from the scientific community to develop molecules able to specifically inhibit the NLRP3 inflammasome. Since the NLRP3 inflammasome pathway includes several steps, a wide range of targets can be used to develop inhibitory strategies. Different inhibitor targets include NLRP3–NLRP3 or NLRP3–ASC interactions, ATP-binding domain blockade with loss of ATPase activity, blockade of NLRP3 posttranslational modification, caspase-1 inhibition, NT-GSDMD pore-formation inhibition, and neutralization of IL-1β and IL-18.

### 4.1. NLRP3 Inhibitors

The advantage of targeting the NLRP3 inflammasome core components is to prevent pyroptosis, an effect that is not affected by IL-β or IL-18 inhibition. The efficacy of NLRP3 inflammasome inhibitors has been studied in vitro, using cellular assays often based on co-stimulation of cells with the lipopolysaccharide (LPS) and ATP (or nigericin, cholesterol crystals, and monosodium urate (MSU) crystals) and/or in vivo. Most of the compounds reviewed here have been tested in animal models of cardiovascular disease. Figure 3 shows some of the inflammasome inhibitors that have been so far tested in the cardiovascular system.

#### 4.1.1. Glyburide, 16673-34-0, and JC-124

Glyburide (or glibencamide) is a sulfonylurea approved for the treatment of type II diabetes [77]. It promotes insulin release, blocking the ATP-sensitive potassium channel (KATP) in pancreatic beta cells [78]. Glyburide was the first identified chemical compound to inhibit the NLRP3 inflammasome in bone marrow-derived macrophages stimulated with LPS/ATP, with no effects on other inflammasomes (e.g., NLRC4, AIM2, or NLRP1), making glyburide specific for NLRP3 [79]. However, the high doses necessary for its anti-inflammasome properties induce severe hypoglycemia and can limit the use of glyburide as an anti-inflammatory drug in vivo [80]. The cyclohexylurea moiety, which is involved in insulin release, is not necessary for the inhibitory activity of the NLRP3 inflammasome [79]. This important piece of information led to the development of an orally active compound, 16673-34-0, lacking the cyclohexylurea moiety but still able to specifically inhibit the NLRP3 inflammasome without affecting glucose metabolism [79]. The administration of 16673-34-0 was tested in different models of cardiac injury [66,79,80]. In mice after ischemia followed by 24 h reperfusion, 16673-34-0 (100 mg/kg) was able to inhibit caspase-1 activity in the heart as well to reduce cardiac injury measured by infarct size and cardiac troponin I release without affecting the glucose levels [66,80]. 16673-34-0 reduced the infarct size even when administered with a 60-min delay after reperfusion [66]. Novel compounds were developed based on 16673-34-0, including JC-124 (N-Me sulfonamide analog of 16673-34-0) [81]. In mice undergoing ischemia (30 or 75 min) and reperfusion (24 h), 30 mg/kg of JC-124 given intraperitoneally was able to reduce the infarct size and plasma troponin I levels [81].

In a non-reperfused model of ischemia, 16673-34-0 ameliorated cardiac function without reducing infarct size [80]. The administration of 16673-34-0 to mice that underwent ischemia in a preclinical model of donation after circulatory death reduced ischemic damage to the heart and improved post-reanimation cardiac function evaluated in vivo [82]. When tested in non-ischemic cardiac injury models, 16673-34-0 showed the same efficacy [80]. In fact, in mice treated with the cardiotoxic chemotherapic drug doxorubicin, intraperitoneal administration of 16673-34-0 (100 mg/kg) improved cardiac function and reduced interstitial fibrosis [80]. In a mouse model of cardiomyopathy induced by a high-sugar–high-fat diet (Western diet), 16673-34-0 administered in the chow (100 mg/kg) prevented systolic and diastolic dysfunction [83]. In mice with experimental pericarditis, 16673-34-0 reduced pericardial effusion and pericardial thickening [84].

The exact mechanism of action of 16673-34-0 has not been uncovered. The efficacy of 16673-34-0 downstream of multiple stimuli and the inefficacy toward AIM2 and NLRC4 inflammasome formation points to the inhibition of NLRP3, prevention of NLRP3 conformational changes, and/or interaction between NLRP3 and ASC [66].

#### 4.1.2. MCC950

MCC950, also known as CP-456,773 or CRID3, was described for the first time in 2001 with other diarylsulfonylurea-containing compounds for their ability to inhibit IL-1β processing [85]. It was further characterized by Coll et al., showing a potent inhibition of NLRP3 both in vivo and in vitro [86]. This compound is a small-molecule that binds noncovalently to NLRP3, close to the Walker B motif blocking NLRP3 ATPase activity, therefore preventing ASC oligomerization and consequent IL-1β release [86,87]. MCC950 is not able to inhibit NLRP1, NLRC4, or AIM2 [86,87,88]. A recent study showed that the molecular target of diarylsulfonylurea inhibitors is an NACHT domain of NLRP3, and as a consequence, MCC950 fails to efficiently inhibit the cryopyrin-associated periodic syndrome (CAPS) forms of NLRP3 [89].

MCC950 has been studied in a mouse model of atherosclerosis [90]. At a daily dose of 10 mg/kg, MCC950 reduces atherosclerotic plaque development, the expression of adhesion molecules in the plaque, as well as and the number of macrophages in the plaque [90]. MCC950 has been shown to have beneficial effects in both small and large animal models of acute myocardial infarction. In pigs, a 7-day treatment (3 or 6 mg/kg) was able to reduce neutrophil infiltration, to reduce myocardial levels of IL-1β, to reduce infarct size, and to preserve cardiac function [91]. In mice with permanent coronary artery occlusion, MCC950 (10 mg/kg) reduced inflammatory cell infiltration, caspase-1 activation, IL-18 and IL-1β levels, and myocardial fibrosis, improving cardiac remodeling [92]. MCC950 seems to be effective in reducing inflammation also when delivered in the ischemic area through hydrolytic microspheres [93]. MCC950 also improved the neurologic function and survival of mice subjected to potassium-based murine cardiac arrest and cardiopulmonary resuscitation [94].

MCC950 (10 mg/kg) administered three times in angiotensin II infusion-induced hypertension has been shown to reduce myocardial fibrosis and IL-1β levels [75]. Twenty-five days of MCC950 treatment (10 mg/kg) can reduce blood pressure and limit renal inflammation in mice with established hypertension. In high fat, high cholesterol, and AngII-treated mice, MCC950 significantly inhibited challenge-induced aortic dilatation, dissection, and rupture in thoracic and abdominal aortic segments [95].

In a mouse model of postmenopausal heart disease, the administration of MCC950 in mice for 8 weeks three times a week (10 mg/kg) limited hypertrophic remodeling and improved systolic and diastolic function and reduced atrial natriuretic peptide (ANP) and BNP mRNA levels [96]. Mice with a cardiomyocyte-specific expression of a mutant form of NLRP3, that is constitutively active, spontaneously developed premature atrial contractions and inducible atrial fibrillation, which was attenuated by MCC950 [19]. The long-term use of MCC950 (20 mg/kg/daily) was able to resemble the protective effects observed in NLRP3^−/−^ mice on cardiac tissues in obesogenic mice (high-fat diet, high-sugar diet, and high-fat–high-sugar diet-fed mice). In these mice, MCC950 given for 15 weeks improved autophagy flux and reduced apoptosis in the heart [97]. MCC950 reduced the myocardial remodeling induced by the infusion of a hypertensive dose of AngII [75].

#### 4.1.3. Bay 11-7082

Bay 11-7082 is a phenyl vinyl sulfone originally identified as a NF-κB pathway inhibitor through blockade of the inhibitor of kappa B kinase (IKK) β [63]. Bay 11-7082 could inhibit the NLRP3 inflammasome with no effects on other inflammasomes tested (NLRP1 and NLRC4). NLRP3 inhibition with Bay 11-7082 is, at least in part, independent of NF-κB-mediated priming inhibition. Bay 11-7082 alkylates cysteine residues in the NLRP3 ATPase region [63].

In a murine model of ischemia/reperfusion, Bay 11-7082 given intraperitoneally 10 min before reperfusion was able to reduce inflammatory cell infiltration as well cardiomyocyte apoptosis and infarct size [98]. In a rat myocardial ischemia-reperfusion model, pretreatment with Bay 11-7082 preserved cardiac function and reduced infarct size as well as cardiac fibrosis and apoptosis [99]. Similar effects were observed in diabetic rats, in which Bay 11-7082 was able to attenuate myocardial injury following ischemia-reperfusion by reducing pyroptotic cell death and NLRP3 inflammasome activation as well caspase-1 and IL-1β expression [44]. However, in vivo, it is difficult to separate the effects of Bay 11-7082 dependent on inhibition of the NF-κB pathway vs. NLRP3 inhibition.

#### 4.1.4. OLT1177

OLT1177 is an orally active beta-sulfonyl nitrile molecule that specifically inhibits the NLRP3 inflammasome [100]. It reduces the release of IL-18 and IL-1β, with no effects on other inflammasomes (AIM2 and NLRC4). OLT1177 inhibits NLRP3 oligomerization, preventing NLRP3–ASC interaction and activation of the downstream cascade. OLT1177 directly interacts with NLRP3 and blocks its ATPase activity [100]. OLT1177 has been tested in several preclinical models of inflammatory disease and consistently blocks NLRP3 activation [100,101,102,103]. In addition, OLT1177 reduced cytokine release in mononuclear cells isolated from patients with CAPS, in which constitutive active mutants of NLRP3 spontaneously release IL-1β and IL-18 in the absence of tissue damage or infection [100]. In an animal model of myocardial ischemia-reperfusion, OLT1177 reduced the infarct size in a dose-dependent manner and preserved cardiac function 24 h and 7 days after reperfusion [104]. OLT1177 also improved ventricular function in a model of coronary artery permanent occlusion. The same study showed the applicability of OLT1177 in a clinically relevant scenario since it has shown efficacy also when given 60 min after reperfusion [104]. Overall, OLT1177 is a promising candidate for the treatment of NLRP3-related diseases, including heart failure (HF) and AMI.

OLT1177 is in clinical testing. In an open-label phase 2A study in patients with gout, a disease dependent on NLRP3 inflammasome activation, OLT1177 was safe and effective in reducing target joint pain [105]. In a pilot phase-1B, double-blind study in patients with heart failure with reduced ejection fraction (HFrEF), OLT1177 was safe and, at the highest dose tested, was associated with an increase in left ventricular ejection fraction and exercise time on a treadmill after 14 days [106].

#### 4.1.5. INF4E

A library of alpha, beta-unsaturated carbonyl– or –cyano derivates was synthesized and screened for their anti-pyroptotic properties [107]. These compounds inhibited NLRP3 through their reactive Michael acceptor moiety. Between these compounds, ethyl 2-((2-chlorophenyl)(hydroxy)methyl)acrylate, INF4E, was chosen for its ability to inhibit the NLRP3 ATPase activity and the activation of caspase-1 [107]. In ex-vivo experiments, pretreatment with INF4E reduced infarct size, decreased lactate dehydrogenase, and improved left ventricular pressure in rat hearts perfused on a Langendorff and subjected to 30 min of ischemia followed by 20- or 60-min reperfusion. Furthermore, INF4E treatment in these hearts reduced the expression of NLRP3 complex components as well as activated the protective reperfusion injury salvage kinase (RISK) pathway and improved mitochondrial function [108]. Due to the potential cytotoxicity of this compound, the same group, in a more recent work, developed other compounds that share the Michael acceptor moiety of INF4E and a sulfonamide or a sulfonylurea portion [109]. The most promising of these compounds is INF58, but its effectiveness as a cardio-protectant has yet to be tested.

#### 4.1.6. Tranilast

Tranilast (N-[3′-4′-dimethoxycinnamonyl]-anthranilic acid, TR), an analog of a tryptophan metabolite, is clinically approved for the treatment of several allergic disorders [110]. It has been shown to reduce collagen synthesis, but the mechanism associated with this function is unknown. Tranilast has recently been identified as a NLRP3 inflammasome inhibitor without noted effects on NLRC4 or AIM2 [111]. It has been shown to directly bind to the NACHT domain of NLRP3, thus inhibiting its ability to oligomerize independently from NLRP3 ATPase activity. Its inhibitory effect is also independent of upstream signaling such as ROS production, ions efflux, or mitochondrial damage [111]. Tranilast showed beneficial pharmacological effects on NLRP3 inflammasome-associated diseases in animal models of CAPS, type 2 diabetes, and gout). More recently, in two mouse models of atherosclerosis, Tranilast promoted NLRP3 ubiquitination, limiting NLRP3 inflammasome assembly and thus resulting in a blunted initiation and progression of atherosclerotic plaques [112]. Before being identified as a NLRP3 inhibitor, Tranilast showed beneficial effects in cardiac fibrosis and remodeling in several animal models of hypertension, diabetic cardiomyopathy and myocardial infarction [113,114,115]. Tranilast has also been used in several clinical trials in which it was shown to be safe and well-tolerated at high doses in patients [116].

#### 4.1.7. CY-09

A recently developed NLRP3 inhibitor, CY-09, binds directly to the ATP-binding motif of the NACHT domain, thus inhibiting NLRP3 assembly and ATPase activity. Its therapeutic efficacy has been tested on animal models of CAPS and type 2 diabetes [117]. The use of CY-09 in a mouse model of diabetic stroke was able to protect from cardiac dysfunction associated with diabetic ischemic stroke [118].

#### 4.1.8. Colchicine

Colchicine is a tricyclic alkaloid already approved for the treatment of inflammatory disorders such as gout and familial Mediterranean fever and is used off label to treat acute and recurrent pericarditis [119]. It works by disrupting microtubule organization and polymerization, thus inhibiting neutrophil chemotaxis and leukocyte diapedesis [120]. However, colchicine has more recently been shown to block NLRP3 inflammasome formation on two levels: preventing P2X7-mediated pore formation and inhibiting intracellular transportation and the spatial arrangement of NLRP3 and ASC, which are subsequently unable to oligomerize [121,122]. In a mouse model of cardiac permanent ligation, colchicine given at 1 mg/kg/day for 7 days improved survival and preserved left ventricular ejection fraction at 4 weeks after surgery [123]. It also reduced the infiltration of neutrophils and macrophages as well mRNA expression of pro-inflammatory cytokines and NLRP3 inflammasome components 24 h after myocardial infarction [123]. Furthermore, colchicine at a low dose (0.5 mg/day) has been shown to be safe and effective in several clinical trials. This dose of colchicine was administered to patients after acute myocardial infarction (COLCOT, COLchicine Cardiovascular Outcomes Trial study, NTC02551094) and significantly reduced the risk of ischemic cardiovascular events compared to placebo at 22 months follow-up [124]. The same dose of colchicine suggests a potential benefit in patients with coronary diseases. In the LoDoCo (Low-Dose Colchicine) study, patients with stable coronary disease receiving colchicine in addition to statins and secondary prevention therapies are at lower risks of cardiovascular events [125]. Colchicine was able to modify coronary plaques and to reduce high-sensitivity C-reactive protein (hsCRP) in patients with post-acute coronary syndrome [126]. In a mouse model of experimental acute pericarditis, colchicine reduced pericardial effusion and the expression of ASC in the pericardium [83].

#### 4.1.9. Hydrogen Sulfide

Hydrogen sulfide (H_2_S) is a gasotransmitter that exerts important physiological functions [127]. It has been shown that H_2_S plays an important role in the response to myocardial ischemia and that H_2_S donors reduce myocardial damage in experimental models of AMI [128]. H_2_S exerts several physiological functions and is cardioprotective due to antioxidative, antiapoptotic, and anti-inflammatory properties [128]. H_2_S is therefore a molecule that has a wide spectrum of activity. The H_2_S donor Na_2_S reduced NLRP3-dependent caspase-1 activation and cell death in primary cardiomyocytes [129]. Na_2_S reduced the infarct size and the caspase-1 activity in mice undergoing ischemia-reperfusion injury. Myocardial protection in vivo and in vitro was dependent on the presence of the microRNA 21. However, a study on macrophages has shown that H_2_S donors sodium thiosulfate or GYY4137 inhibit signaling leading to NLRP3 inflammasome activation [130]. NaHS, another H_2_S donor, reduced the IKKβ/NF-κB signaling pathway and was cardioprotective in a model of hemorrhagic shock [131]. Thus, H_2_S can reduce inflammasome activity by acting on priming and trigger signaling.

## 5. Inhibition of Caspase-1, IL-1, and IL-18

Blockade of the inflammasome components ASC and caspase-1 or the inflammasome products IL-1β and IL-18 can produce overlapping effects with NLRP3 inhibition. However, because those components and products are not unique to the NLRP3 inflammasome, their blockade may interfere with the activity of other inflammasomes.

A few ASC inhibitors have been developed but not tested in models of cardiovascular disease. In in vitro or ex vivo ischemic models, caspase-1 inhibition improved cardiomyocyte contractility and reduced damage [132,133,134].

Several IL-1 inhibitors have been used in clinical practice for several years, although they are not approved to treat cardiovascular diseases. As stated above, IL-1 activity promotes the development and the instability of atherosclerotic plaques. In fact, in mice, deletion of the IL-1RI receptor or IL-1α and IL-1β reduces the size of the plaque [53]. IL-1α has a role in the early phases of early experimental atherogenesis and IL-1β promotes inflammation and plaque remodeling in the late phases of atherosclerosis [135]. To prove the central role of IL-1β in established atherosclerotic disease, in the Canakinumab Anti-Inflammatory Thrombosis Outcomes Study (CANTOS), canakinumab, a monoclonal antibody that specifically inhibits IL-1β, reduced the number of atherothrombotic events [136].

Inhibition of IL-1 signaling using anakinra, a recombinant form of the human IL-1Ra, or IL-1 Trap, a chimeric protein that inhibits IL-1α and β, in a mouse model of AMI due to permanent coronary artery occlusion reduced the adverse ventricular remodeling [137,138]. Given before ischemia-reperfusion, anakinra reduced the size of the infarct and improved the ventricular function of the mouse heart [139]. Studies performed in mice to define the role of IL-1α or β in ischemia-reperfusion injury showed that a blocking antibody developed against IL-1α reduced the infarct size while a blocking antibody developed against IL-1β had no effect on the size of the infarct [140,141]. However, in the mouse model of AMI induced by permanent coronary artery occlusion, IL-1β blockade with one of two different monoclonal antibodies reduced adverse ventricular remodeling and improved myocardial contractility [142,143,144]. The VCUART (Virginia Commonwealth University Anakinra Remodeling Trials) was three sequential double-blinded placebo-controlled phase II clinical studies performed in a population of patients with ST segment elevation AMI [145,146,147]. These studies showed that, compared to the placebo group, the patients treated with anakinra had reduced levels of C-reactive protein (a marker of inflammation). Furthermore, compared to the placebo-treated group, fewer patients had new onset HF and HF hospitalization in the anakinra group [145,146,147]. The MRC-ILA-Heart study enrolled patients with non-ST-segment elevation myocardial infarction. In this patient population, anakinra reduced acute inflammatory response but failed to improve clinical outcomes [148].

IL-1β reduces myocardial contractility in vitro and in vivo [149]. It reduces myocardial relaxation and beta-adrenergic receptor responsiveness in healthy mice [149]. In hospitalized patients with acute decompensated systolic HF, anakinra reduced acute inflammatory response and improved ejection fraction [150]. In a sub-study of the CANTOS trial, canakinumab reduced the rate of hospitalizations for HF [151]. In the Recently Decompensated Heart Failure Anakinra Response Trial (REDHART), anakinra improved cardiorespiratory fitness (peak oxygen consumption), reduced NT-proBNP levels, and improved the quality of life of the patients [152]. The REDHART2 study is ongoing [153]. In the Diastolic Heart Failure Anakinra Response Trial (D-HART), anakinra was administered to patients with HF with preserved ejection fraction. Anakinra promoted a significant improvement in the patients’ peak oxygen consumption [154]. The D-HART2 study was conducted in the same population as in the first study, although patients had a higher body mass index. The study showed that anakinra promoted a significant increase in treadmill exercise time, lowered the NT-proBNP levels, and improved quality of life in the absence of a significant change in cardiorespiratory fitness [155].

IL-18 has important effects on the cardiovascular system as well [156]. Following AMI, the systemic IL-18 levels increased and predicted a worse outcome [157,158]. IL-18 levels also increased with the severity of HF [159]. Mice pretreated with an antibody that neutralized IL-18 before ischemia-reperfusion injury had a smaller infarct size than the controls [160]. Treatment with a recombinant IL-18 binding protein (IL-18BP) mitigated damage to the heart and inflammation in a mouse model of heterotopic heart transplantation [161]. In a mouse model of heart donation after circulatory death, the hearts that were reanimated ex vivo in the presence of IL-18BP had improved contractility and reduced markers of myocardial damage [162]. In a model of in vitro ischemia, human myocardial strips that were incubated with IL-18BP had better contractility than the controls [131]. In a mouse model of chronic alveolar hypoxia, IL-18BP improved the function of the right ventricle [163]. Finally, an antibody against IL-18, given to mice with myocardial injury caused by β-adrenergic receptor overstimulation, reduced heart damage, decreased fibrosis, and improved myocardial function [164].

## 6. Conclusions

The NLRP3 inflammasome and the cytokines it regulates are increased in biological samples of patients with different types of heart or vascular diseases. Experimental models of these diseases have proven that the NLRP3 inflammasome and its products have a central role in the pathogenesis of heart diseases as well as atherosclerosis. The development of specific NLRP3 inhibitors is proving to be a successful strategy in reducing myocardial injury, in preventing adverse myocardial remodeling, and in improving heart contractility. The clinical data collected using colchicine, OLT1177, and IL-1 inhibitors are very encouraging and point out that NLRP3 inhibition is a feasible and effective strategy. In addition to the inhibitors described here, other NLRP3-specific inhibitors (reviewed elsewhere) have been developed but were not described here due to the lack of data in the cardiovascular models of disease [165]. In conclusion, animal models have demonstrated very robust evidence of a protective effect of NLRP3 inflammasome inhibition and the results of early clinical trials that aimed to limit the effects of the NLRP3 inflammasome in cardiovascular diseases are promising.

## Figures and Tables

**Figure 1 molecules-26-00976-f001:**
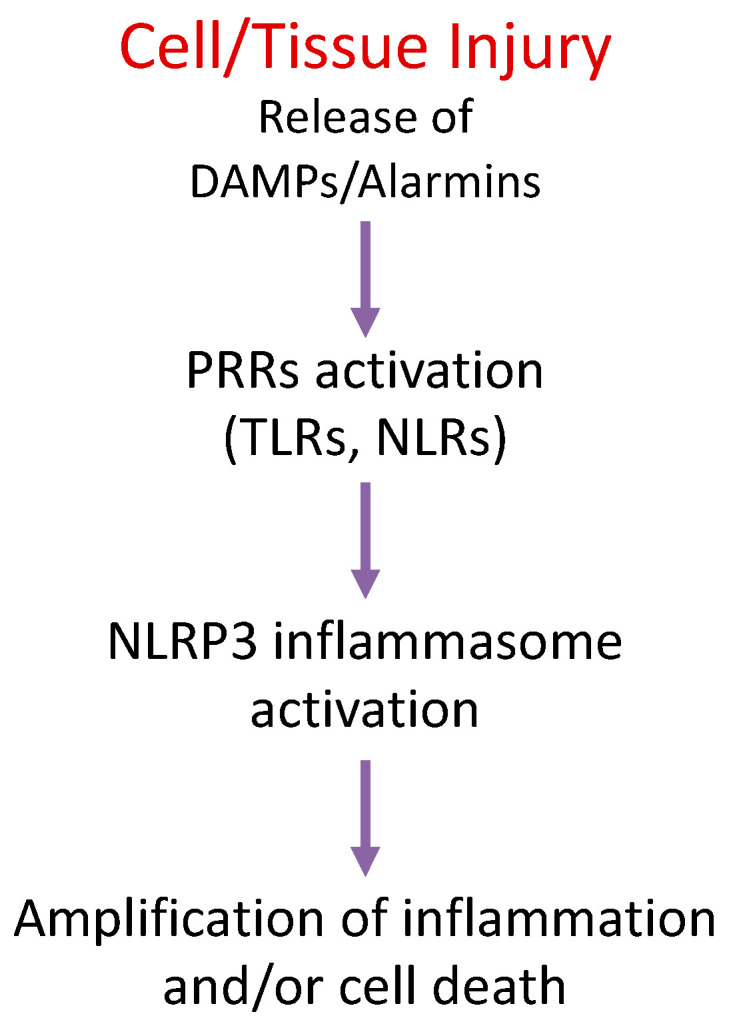
Schematic representation of the critical steps that lead from cellular injury to the inflammatory response. DAMPs, damage associated molecular patterns; PRRs, pattern recognition receptors; TLRs, Toll-like receptors; NLRs, Nod-like receptors; NLRP3, NACHT, leucine-rich repeat (LRR), and pyrin domain (PYD)-containing protein 3.

**Figure 2 molecules-26-00976-f002:**
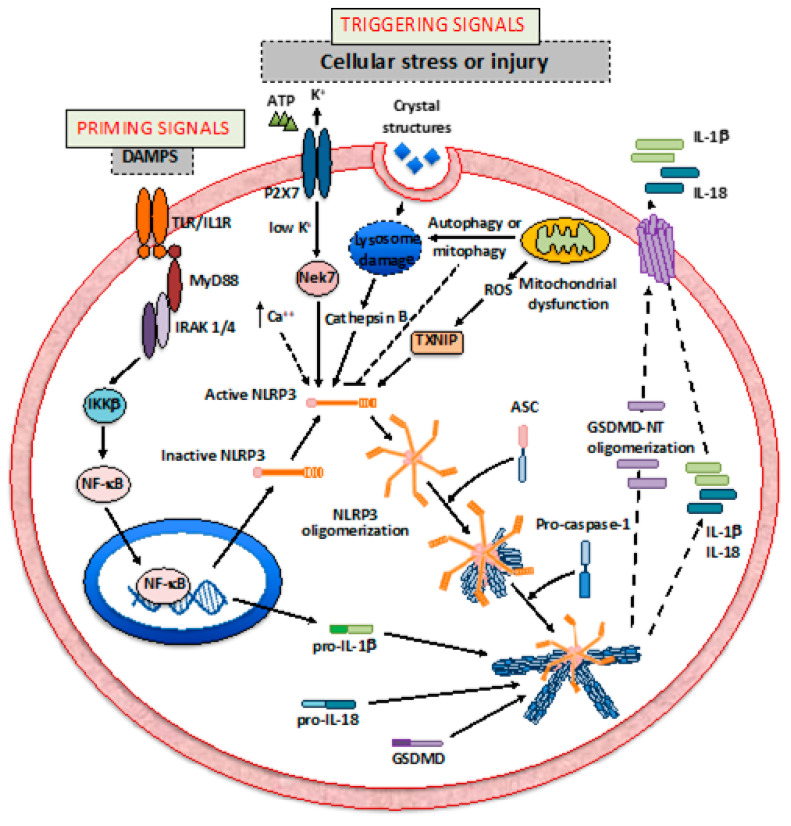
The priming and triggering signals that regulate the inflammasome are initiated by the cellular stressors or by injury. The priming ones regulate transcription of the inflammasome components mainly though nuclear factor kappa B (NF-κB) activity, while the triggering ones consist of the signal necessary to activate NLRP3, leading to the assembly of the inflammasome; activation of caspase-1; and cleavage of the substrates pro-interleukin (IL)-1β, pro-IL-18 and Gasdermin D (GSDMD). The N-terminal (NT) fragment of GSDMD forms pores that allow the secretion of active IL-1β and IL-18. DAMPs, damage-associated molecular patterns; TLRs, Toll-like receptors; IL-1R, interleukin-1 receptor; MyD88, myeloid differentiation factor 88; IRAK1/4, interleukin-1 receptor-associated kinase 1 and 4; IKKβ, inhibitor of *kappa* B kinase β; NF-κB, nuclear factor *kappa* B; NLRs, Nod-like receptors; Nek7, NIMA-related kinase 7; P2X7, Purinergic receptor 2 X 7; ROS, reactive oxygen species; TXNIP, thioredoxin interacting protein; NLRP3, NACHT, leucine-rich repeat (LRR), and pyrin domain (PYD)-containing protein 3; ASC, apoptosis-associated speck-like protein containing a caspase recruitment domain.

**Figure 3 molecules-26-00976-f003:**
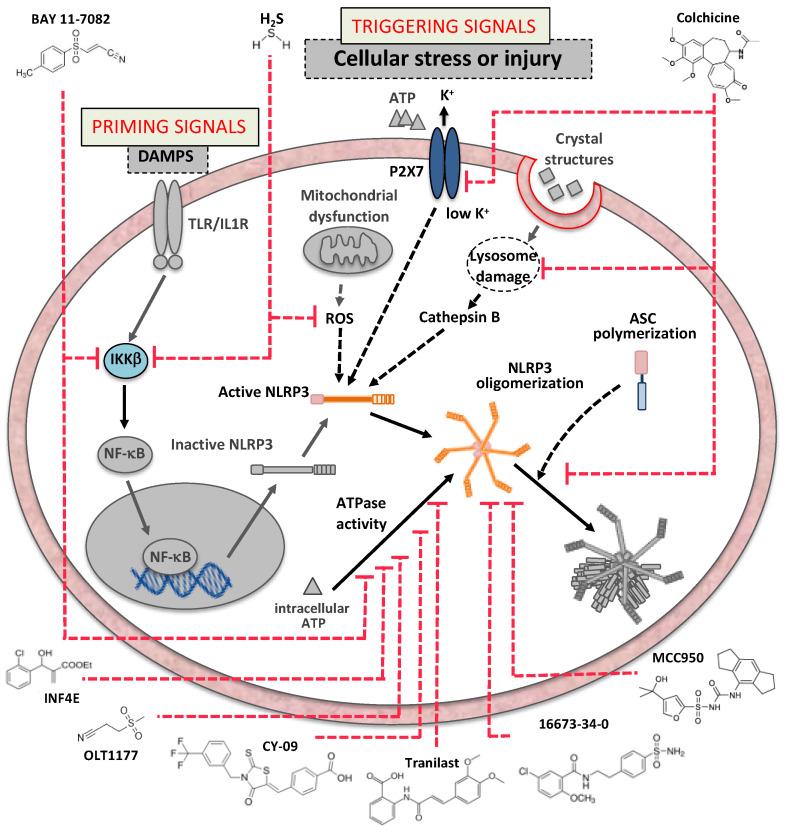
Inflammasome inhibitors tested in the cardiovascular system are reported together with the molecular target and mechanism of action. DAMPs, damage-associated molecular patterns; TLRs, Toll-like receptors; IL-1R, interleukin-1 receptor; IKKβ, inhibitor of *kappa* B kinase β; NF-κB, nuclear factor *kappa* B; P2X7, Purinergic receptor 2 X 7; ROS, reactive oxygen species; NLRP3, NACHT, leucine-rich repeat (LRR), and pyrin domain (PYD)-containing protein 3; ASC, apoptosis-associated speck-like protein containing a caspase recruitment domain.

## Data Availability

Not applicable.

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
