# Peer review of "NLRP3 Inflammasome Inhibitors in Cardiovascular Diseases"

_molecules, 2021, doi:10.3390/molecules26040976_

Round 1

Reviewer 1 Report

This review evidences the deleterious role that the NLRP3 inflammasome has on cardiovascular disease, with a sound mechanistic discussion. The review includes a diverse range of nine small molecule inflammasome inhibitors from different mechanistic classes, which have been tested in animal models of cardiovascular disease, including acute myocardial infarction; reduction of risk and/or cardiac injury was observed, especially in the promising clinical studies with Colchicine and OLT1177. The impact of biologicals on cardiovascular disease, including anakinra, has also been detailed. The review is well illustrated and also has a comprehensive reference list. This manuscript is suitable for publication, subject to addressing the minor suggestions/corrections below.

Line 15: has opened the (delete to)

Line 25: bind to the (add to)

Line 30: in the absence (add the)

Line 32: that when released (delete are)

Line 34: or alternatively

Line 131: other types

Lines 210-213, Delete along with all the references: ‘A pathophysiological role of the NLRP3 inflammasome has been also shown in animal models of injury due to anti-cancer treatments (i.e. chemotherapy and radiation therapy), obesity and age-associated metabolic derangements, diabetic cardiomyopathy, pericarditis, myocarditis and cardiac sarcoidosis [77-102].’ OR pick out relevant key studies to discuss?

Line 219: or an

Line 221: reviewed here (delete manuscript)

Line 229: Glyburide was the first….

Line 223, Figure 3: Improve sharpness of structure of 16673-34-0

Line 232: no effects shown….

Line 245: JC-124 (N-Me sulfonamide analogue of 16673-34-0)

Line 261: points to

Line 278: day

Line 356-358: Sentence reword please, and spelling of Michael.

Line 367: oligomerize

Line 430: inhibitors have been used…

Author Response

We thank the reviewer for the detailed revision of the manuscript. 

in the revised version of the manuscript, all the concerns of the reviewer have been addressed.

We decided to keep the statement from line 210 to line 213 and change the reference to two review articles. We added a statement to justify the presence of the statement.

We hope that the current version of the manuscript meets the expectations of the reviewer.

Reviewer 2 Report

When asked to evaluate the review article written by Mezzaroma, Abbatte, and Toldo on Christmas day, my immediate reaction was to not accept the invitation due to the timing and the fact that there are already too many inflammasome reviews in the literature. Nevertheless, I accepted the invitation and was very pleasantly surprised by the overall quality, clarity, and conciseness of the manuscript. I was able to follow the intent of the narrative, and I feel that its contents deserve to be published and that they will add valuable insight to the complement the existing literature. The citations are both classical and contemporary, and allow the reader to access the relevant primary papers. My comments are largely those of an editorial nature, which are provided to increase the readability of this otherwise already-excellent manuscript. I have mentioned these in the order they appear in the manuscript, by line, although a few of these are much more important than others as they require rewriting to clarify the authors’ intents or to more factually convey the information.

Title:    Add “NLRP3” to the beginning of the title since this inflammasome if the exclusive focus of the review from the perspective of inhibitors used in the pre-clinical and clinical studies cited.

32:       delete “are”

37:       change “protein” to “proteins”

42:       add “infectious” to “rheumatologic and chronic diseases”

44:       change “like” to “such as”

47:       add “injury” after “cellular:”

49:       delete “the”

50:       change “like” to “including”

53:       change “protein” to “proteins”

56:       change “k” to a kappa symbol

57:       change “like” to “such as”; change “As anticipated above, NLRP3” to “NLRP3 has been”

59:       add ”to be” between ”but” and “less”

68:       describe the consequences of AIM2 activation by adding one or a few sentences

71:       define LRRs (the definition appears later, at line 85)

77:       change “k” to a kappa symbol

79:       add that TLR activation also promotes transcription and translation of the caspase-1 substrates, pro IL-1beta and pro-IL-18

91:       place this line above Figure 2

Figure 2:          use beta symbols for IL-1b (3x) and IKKb (1x), and use kappa symbols for NF-kB (2x)

94:       change “k” to a kappa symbol

111:     add a comma after “oligomerizes”

112:     delete “are”

113:     add “are” after “cytokines”

116:     add a comma after “pore”

117:     add a comma after “swelling”

119:     change “to” to “for”

120:     change “In” to “Under”

121:     add a comma after “components”

128:     change “cytokines” to “cytokine”

129:     change “k” to a kappa symbol

131:     change ”type” to “types”

132:     delete “the” before “cardiovascular”

134:     delete “the”

136:     change “like” to “such as”; add a comma after “phosphorylation”

143:     add “preceding inflammasome activation” after “phase”

147:     change “the” to “inflammasome”

150:     add a comma after “inflammasome”

153:     delete “the” before “K+

160:     add a comma after “phosphate”

161:     add commas after “instability” and “swelling”

178-80:            This sentence is unclear

182:     change “promptly” to ”prompt”

187:     add a comma after “site”

192:     add a comma after “failure)”

195:     delete “the”

203:     add a comma after “myocarditis”

211:     should “ATP-activity” be “ATPase-activity”?

213:     add “and” before “neutralization”

223:     move this line above Figure 3

Figure 3:          use symbols in IKKb and NF-kB (2x)

229:     add “to be” between “shown” and “the”

231:     delete “were shown”

238:     delete “the” after “affecting”

246:     change “were” to “was”

248-51:            use past tense, as in other parts of the review

261:     delete “out”

262:     add a comma after “changes”

266:     there is no need to identify authors by name since it isn’t done for the majority of other citations

268-9:  add “thereby” before “, preventing”

269:     change “It” to “MCC950”

270:     there is no need to identify authors by name since it isn’t done for the majority of other citations

272:     add a comma after “NLRP3”; delete the comma after “consequence”

276:     add a comma after the first “plaque”

279:     delete “to”; add a coma after “size”

280:     move comma after “mice” to after “occlusion”

280-2:  make into past tense

287:     delete “for”

288:     delete comma after “hypertension”

291:     add a comma after “mice”

291-2:  add a comma after “dissection”

294:     add a comma after “10 mg/kg)”

302:     change “reduces” to “reduced” (2x)

305:     change the “k” in “kB” to a kappa symbol

306:     there is no need to identify authors by name since it isn’t done for the majority of other citations

307:     change “can” to “could”

309:     change the “k” in “kB” to a kappa symbol

313:     change “cardiomyocytes” to “cardiomyocyte”

314:     delete comma after “11-7082”

316:     change “where” to “in which”

319:     change the “k” in “kB” to a kappa symbol and “vs” to “versus”

322:     delete the comma after “OLT1177”

324:     add a comma after “oligomerization”

325:     delete the first “the”

328:     change “it” to “OLT1177”

329:     define the acronym CAPS

332:     add “and” after “manner”

333:     change “improves” to “improved”

337:     define the acronym HF

339”    delete “the”

342:     delete “it”

345:     there is no need to identify authors by name since it isn’t done for the majority of other citations

348:     add “being” before “able”

352:     change “lactate dehydrogenase” to “decreased lactate dehydrogenase,”

356:     change “Doing” to “Due”

359-60:            add “a” before “cardioprotectant”

365:     change “Recently it has” to “Tranilast has recently”

367:     change “oligomerizes” to “oligomerize”

369:     change “like” to “such as”

372:     change “it” to “Tranilast”

377:     change “showing to be safe and well tolerated” to “, in which it was shown to be safe and well-tolerated”

380:     change “assembling” to “assembly”

386:     add a comma after “Fever”

387:     change “microtubules” to “microtubule”

388:     change “leukocytes” to “leukocyte”

389:     change “more recently has” to “colchicine has more recently”; delete “its ability”

391:     add “subsequently” after “are”

399:     add “and” before “significantly”

399-400:          change “reduces” to “reduced”

401:     change “beneficial” to “benefit”

405:     define hsCRP

413:     add a comma after “antiapoptotic”

414-6:  make past tense as deemed by the authors to be appropriate

420:     subscript the “2” in “H2S”; make symbols for IKKbeta and NF-kappaB

421:     add “by” after “activity”

424:     change “IL18” to ”IL-18”

424-6:  clarify that inhibition of ASC or caspase-1, but unlikely IL-1beta or IL-18, could affect other inflammasome pathways in addition to NLRP3

428-9:  make past tense

430:     change ”are” to “have been”

442, 443, and 446:      changes “reduces” to “reduced”

448:     add a comma after “occlusion”

466:     add a comma after “levels”

476:     change “AMl” to ”AMI”

477:     change “increase also” to “also increase”

478:     change “have” to “had”

479:     change “mitigates” to “mitigated:”; define the acronym IL-18BP

481:     add “the” before “presence”

487:     add “decreased the” before “fibrosis”; add a comma after “fibrosis”

490:     change “its cytokines” to ”the cytokines it regulates”

491:     add “these” before “diseases”

495:     add a comma after “remodeling”

502:     add “in cardiovascular diseases” after “inflammasome”

Author Response

The authors are very grateful for the reviewer’s careful and detailed revision of the manuscript. 

We have addressed all the reviewer’s comments and have made all the changes suggested.

All the grammatical changes, the corrections of typos and the conceptual changes have been made accordingly to the reviewer’s suggestion.

Unclear sentences have been revised per reviewer’s suggestion.

We sincerely hope that the reviewer finds the new version of the manuscript suitable for publication.